# *Bifidobacterium lactis* Probio-M8 Adjuvant Treatment Confers Added Benefits to Patients with Coronary Artery Disease via Target Modulation of the Gut-Heart/-Brain Axes

Baoqing Sun,[d] Teng Ma,[a,b,c] Yalin Li,[a,b,c] Ni Yang,[a,b,c] Bohai Li,[a,b,c] Xinfu Zhou,[d] Shuai Guo,[a,b,c] Shukun Zhang,[e] Lai-Yu Kwok,[a,b,c] (ORCID) Zhihong Sun,[a,b,c] Heping Zhang[a,b,c]

[a]Inner Mongolia Key Laboratory of Dairy Biotechnology and Engineering, Inner Mongolia Agricultural University, Hohhot, Inner Mongolia, China
[b]Key Laboratory of Dairy Products Processing, Ministry of Agriculture and Rural Affairs, Inner Mongolia Agricultural University, Hohhot, Inner Mongolia, China
[c]Key Laboratory of Dairy Biotechnology and Engineering, Ministry of Education, Inner Mongolia Agricultural University, Hohhot, Inner Mongolia, China
[d]Department of Cardiology, Weihai Municipal Hospital, Cheeloo College of Medicine, Shandong University, Weihai, Shandong, China
[e]Department of Pathology, Weihai Municipal Hospital, Cheeloo College of Medicine, Shandong University, Weihai, Shandong, China

Baoqing Sun, Teng Ma, and Yalin Li contributed equally to this work.

**ABSTRACT** Accumulating evidence suggests that gut dysbiosis may play a role in cardiovascular problems like coronary artery disease (CAD). Thus, target steering the gut microbiota/metabolome via probiotic administration could be a promising way to protect against CAD. A 6-month randomized, double-blind, placebo-controlled clinical trial was conducted to investigate the added benefits and mechanism of the probiotic strain, *Bifidobacterium lactis* Probio-M8, in alleviating CAD when given together with a conventional regimen. Sixty patients with CAD were randomly divided into a probiotic group ($n = 36$; received Probio-M8, atorvastatin, and metoprolol) and placebo group ($n = 24$; placebo, atorvastatin, and metoprolol). Conventional treatment significantly improved the Seattle Angina Questionnaire (SAQ) scores of the placebo group after the intervention. However, the probiotic group achieved even better SAQ scores at day 180 compared with the placebo group ($P < 0.0001$). Moreover, Probio-M8 treatment was more conducive to alleviating depression and anxiety in patients ($P < 0.0001$ versus the placebo group, day 180), with significantly lower serum levels of interleukin-6 and low-density lipoprotein cholesterol ($P < 0.005$ and $P < 0.001$, respectively). In-depth metagenomic analysis showed that, at day 180, significantly more species-level genome bins (SGBs) of *Bifidobacterium adolescentis*, *Bifidobacterium animalis*, *Bifidobacterium bifidum*, and *Butyricicoccus porcorum* were detected in the probiotic group compared with the placebo group, while the abundances of SGBs representing *Flavonifractor plautii* and *Parabacteroides johnsonii* decreased significantly among the Probio-M8 receivers ($P < 0.05$). Furthermore, significantly more microbial bioactive metabolites (e.g., methylxanthine and malonate) but less trimethylamine-N-oxide and proatherogenic amino acids were detected in the probiotic group than placebo group during/after intervention ($P < 0.05$). Collectively, we showed that coadministering Probio-M8 synergized with a conventional regimen to improve the clinical efficacy in CAD management. The mechanism of the added benefits was likely achieved via probiotic-driven modulation of the host's gut microbiota and metabolome, consequently improving the microbial metabolic potential and serum metabolite profile. This study highlighted the significance of regulating the gut-heart/-brain axes in CAD treatment.

**IMPORTANCE** Despite recent advances in therapeutic strategies and drug treatments (e.g., statins) for coronary artery disease (CAD), CAD-related mortality and morbidity remain high. Active bidirectional interactions between the gut microbiota and the heart implicate that probiotic application could be a novel therapeutic strategy for CAD. This study hypothesized that coadministration of atorvastatin and probiotics

**Ad Hoc Peer Reviewer** Tomasz Wilmanski

Address correspondence to Heping Zhang, hepingdd@vip.sina.com.

The authors declare no conflict of interest.

could synergistically protect against CAD. Our results demonstrated that coadministering Probio-M8 with a conventional regimen offered added benefits to patients with CAD compared with conventional treatment alone. Our findings have provided a wide and integrative view of the pathogenesis and novel management options for CAD and CAD-related diseases.

**KEYWORDS** Coronary artery disease, *Bifidobacterium lactis* Probio-M8, species-level genome bins (SGBs), metabolomics, gut-heart axis, gut-brain axis, gut-heart/brain axes

Coronary artery disease (CAD), also called coronary heart disease (CHD), refers to myocardial dysfunction and/or organic lesions caused by coronary artery stenosis and insufficient blood supply, which is a common cardiovascular disease (CVD) that afflicts 110 million patients (1, 2). Despite recent advances in medicine and therapeutic strategies, the mortality and morbidity associated with CVD remain very high, especially noticeable in high-and intermediate-income countries (3). Statins are widely used as first-line drugs for the treatment of dyslipidemia and atherosclerosis (4). However, serious side effects have been reported in using statins, such as rhabdomyolysis, liver toxicity, and a higher risk for new-onset diabetes (5). Recent studies have also explored the role of gut microbiota and its derived metabolites (such as short-chain fatty acids [SCFAs], trimethylamine-N-oxide [TMAO], bile acids) in the pathogenesis of CAD (6), and a large body of evidence supports that the gut microbiota composition and its activity might play a role in atherosclerosis (7). The "gut-heart axis" via bidirectional communications between the gut and the heart together achieve cardiac homeostasis, and recent studies have shed new insight into the intricate mechanisms of CAD (8). Active bidirectional interactions between the gut microbiome and the heart implicate that applying products like prebiotics, postbiotics, and probiotics to target the gut microbiota and its metabolites would be an interesting and novel therapeutic approach for preventing and managing CAD (9, 10).

Probiotics are defined as "live microorganisms that confer a health benefit on the host when administered in adequate amounts" (11). Increasing evidence supports that probiotics can alleviate disorders related to the immune system, depression, anxiety, type 2 diabetes, obesity, and gastrointestinal and cardiovascular health (12, 13). *Lactobacillus* (*L.*) and *Bifidobacterium* (*B.*) are the most used probiotic bacteria. Results from animal models indicate that certain probiotic strains may have protective and alleviating effects on CAD. Rats treated with supplements containing *L. plantarum* 299v before coronary artery ligation have reduced infarct size with improvement in left ventricular function (14). Administering *Akkermansia muciniphila* improved inflammation and reduced atherosclerotic lesions by restoring the intestinal barrier in hypercholesterolemic *Apo*E$^{-/-}$ mice (15). Another study found that ingesting *L. coryneformis* CECT5711 improved endothelium-dependent vasodilation and reduced vascular oxidative stress in obese mice, suggesting target manipulation of intestinal microbes could potentially help prevent CVD (16). In a human clinical trial, the consumption of *L. reuteri* V3401 for 12 weeks effectively reduced the risk for CVD in obese adults, with a reduction in the levels of multiple inflammatory biomarkers (such as TNF-$\alpha$, IL-6, and IL-8) (17). A randomized, double-blind, placebo-controlled trial showed that 12-week probiotic supplementation was beneficial to diabetic patients with CAD (18), as reflected by the changes in their glycemic control, levels of high-density lipoprotein cholesterol (HDL-C), biomarkers of inflammation, and oxidative stress. These findings supported that the administration of exogenous probiotics might be a promising way to improve the therapeutic effect of CAD. In another clinical trial, 70 pregnant women who consumed probiotic yogurt containing *B. lactis* Bb12 for 9 weeks had significantly reduced serum levels of total cholesterol and triglycerides (19). *Bifidobacterium lactis* Probio-M8 (Probio-M8) is a probiotic strain isolated from the human breast milk of a healthy woman, and it has been shown to confer various beneficial effects to the host (20, 21). As Probio-M8 belongs to the same subspecies as the *B. lactis* Bb12 strain, similar hypolipidemic effects would be

anticipated. However, as the functional properties of probiotics are strain-specific, it would thus be necessary to evaluate the therapeutic efficacy of each strain independently.

Both statins and probiotics are efficacious in improving CAD-associated symptoms, but only limited studies have investigated the combined effects of the two. Therefore, in this study, we hypothesized that coadministration of atorvastatin and probiotics could synergize the protective effect against CAD. The main objective of the current study was to conduct a randomized, double-blind, placebo-controlled trial to investigate the synergistic effect of the conventional drug treatment with probiotics (received Probio-M8 powder, atorvastatin, and metoprolol) in comparison with the placebo group without receiving probiotics (only given placebo powder, atorvastatin, and metoprolol). The primary outcomes were various clinical indicators for coronary artery disease, namely, angina frequency (AF), angina stability (AS), disease perception, physical limitation (PL), treatment satisfaction (TS), anxiety and depression levels (evaluated by the scores on the Self-Rating Anxiety Scale scores [SAS], and Self-Rating Depression Scale scores [SDS]); serum indicators, including interleukin-6 (IL-6), low-density lipoprotein cholesterol (LDL-C), cereal third transaminase, blood urea nitrogen, and creatinine; and white blood cell count. Additionally, changes in patients' fecal metagenomes and serum marker metabolites were followed. The second objective of this study was to explore the possible mechanisms behind the symptom-attenuating effects. The findings of the present study would provide a wide and integrative view of the pathogenesis and reference information for managing CAD, expanding the knowledge and application of probiotics-based therapy.

## RESULTS

**Adjunctive probiotics improved CAD-related symptoms.** After the intervention (at day 180), the SAQ scores increased significantly, while the SAS and SDS scores and the serum levels of LDL-C and IL-6 decreased significantly in both Probio-M8 and placebo groups ($P < 0.01$ in all cases; Fig. 1B), suggesting that the intervention with atorvastatin and metoprolol both with and without adjunctive probiotics could effectively alleviate the clinical symptoms of CAD patients. However, interestingly, treatment of probiotics in conjunction with conventional drugs resulted in a significantly larger magnitude of increase in SAQ scores ($P < 0.0001$), and the subscores of AF, AS, PL, and TS increased by 1.14-fold, 1.13-fold, 1.17-fold, and 1.08-fold, respectively, compared with the placebo group. Moreover, Probio-M8 supplementation augmented the anti-depression and anti-anxiety effects (SAS and SDS; $P < 0.0001$ versus placebo group; Fig. 1B). Moreover, coadministering Probio-M8, atorvastatin, and metoprolol continuously for 6 months significantly reduced the levels of IL-6 and LDL-C ($P < 0.005$; Fig. 1B), and such effects were stronger in the Probio-M8 group compared with the placebo group.

On the other hand, no significant difference was observed in the levels of cereal third transaminase (a liver function indicator), blood urea nitrogen (a renal function indicator), creatinine, and white blood cell count between the two groups (Table S3). Additionally, during the 6-month follow-up period, the incidence of adverse effects was significantly lower in the Probio-M8 group (15.7%) than the placebo group (25%; $P < 0.05$). These results demonstrated that the application of Probio-M8 as adjunctive treatment significantly improved CAD-associated clinical symptoms and reduced the occurrence of adverse events.

**Genomic characteristics of the gut microbiome in patients with CAD.** In-depth metagenomic analysis was conducted on a total of 123 samples from 41 participants collected at days 0, 90, and 180. A total of 10,977 metagenome-assembled genomes (MAG) were obtained by bin refinement module using MetaBAT2, VAMB, and DAS tools, and 2,972, 569, and 654 MAGs were assigned to high-, medium-, and partial-quality MAGs, respectively. A total of 440 SGBs were extracted from these high-quality MAGs (Table S4; distributed across 13 phyla, 20 classes, 26 orders, 41 families, 83 genera, and 281 species), while most SGBs were assigned to the phylum Firmicutes (70.68%), followed by Bacteroidetes (11.59%), Actinobacteria (6.59%), and Proteobacteria (7.95%).

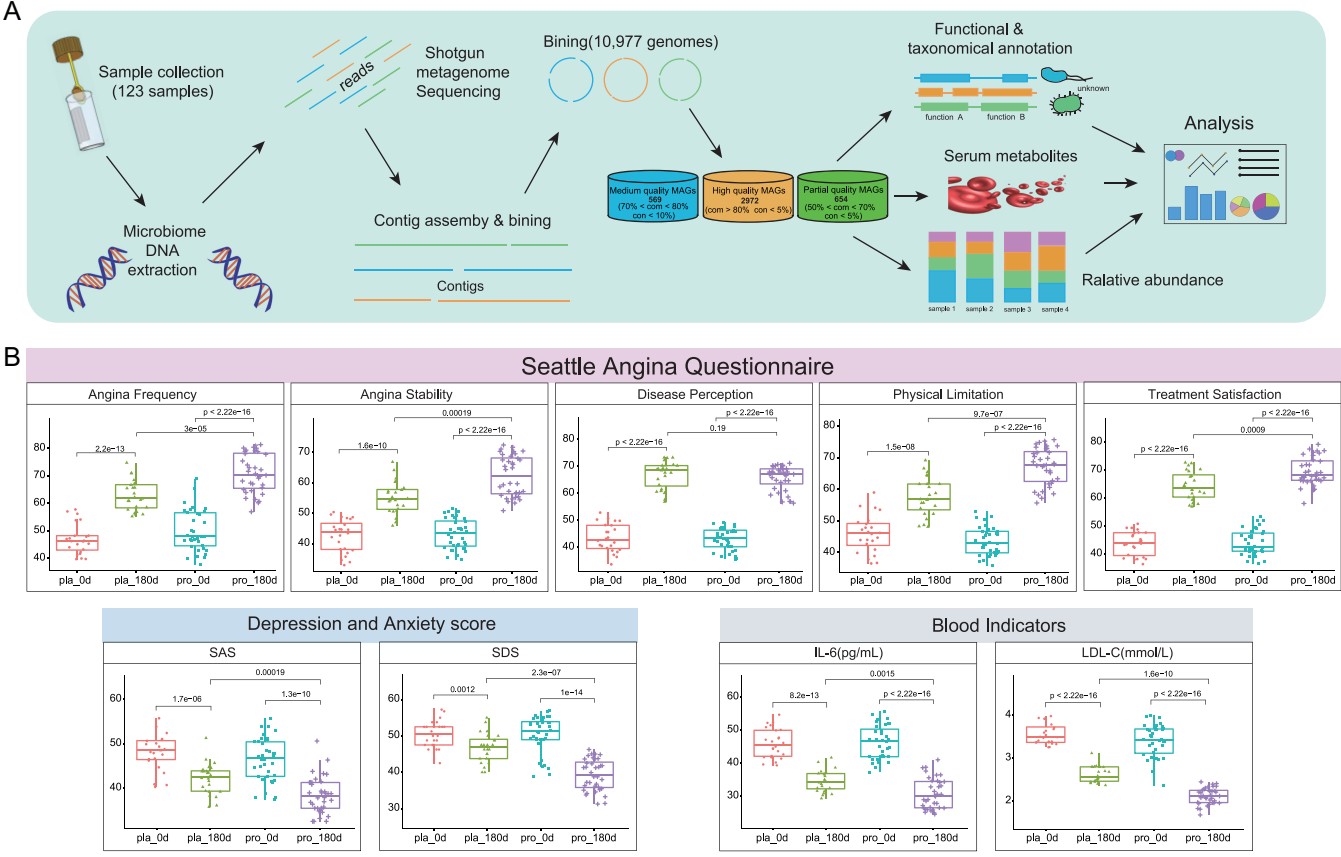

**FIG 1** Clinical indicators of coronary artery disease-associated symptoms and multiomics analysis pipeline. (A) The workflow of microbial community composition, functional taxonomical annotation, and serum metabolome multiomics analysis. A total of 2,972 high-quality metagenome-assembled genomes (MAGs) were identified in the complete data set. "com" and "con" represent levels of completeness and contamination, respectively. Statistical differences were analyzed using the Wilcoxon test or $t$ test. (B) Statistical differences in clinical indicators were evaluated by the Wilcoxon test (for horizontal comparison between probiotic and placebo groups; $n = 36$ and 24, respectively) or paired $t$ test (for longitudinal comparison between days 0 and 180), respectively. Benjamini-Hochberg procedure was applied to correct for multiple testing in all cases and corrected $P < 0.05$ was considered statistically significant. Abbreviations: SAS=Self-Rating Anxiety Scale scores; SDS=Self-Rating Depression Scale scores; IL-6 = interleukin-6; LDL-C = low-density lipoprotein cholesterol.

**Adjunctive probiotics administration modulated patients' gut microbiota composition.** To assess the effect of probiotic consumption on the intestinal microbiome of patients with CAD, changes in the alpha (represented by the Shannon diversity index) and beta diversity (evaluated by principal coordinates analysis (PCoA), Bray-Curtis distance) were monitored. No significant fluctuations were observed in the Shannon diversity index for both groups during the intervention. Similarly, the PCoA analysis found no significant difference in the overall gut microbiota structure between different time points in both groups ($P > 0.05$; Fig. 2A and B). Although no significant change was detected in the overall gut microbiome diversity in both groups, dramatic changes were observed in the abundance of some crucial SGBs after Probio-M8 administration. These were responsive SGBs that did not show a significant difference in abundance between Probio-M8 and placebo groups at day 0 but became differentially abundant between time points or between treatment groups (Table S5). In the Probio-M8 group, eight responsive SGBs significantly increased in abundance, including *B. adolescentis* (A073.M001), *B. animalis* (A222.M001), *B. bifidum* (A061.M008), and *Ruminococcaceae* bacterium (A273.M017), while *Roseburia inulinivorans* (B143.M007), and *Eubacterium* sp. CAG:115 (B201.M002) decreased significantly during/after the intervention ($P < 0.05$). On the other hand, surprisingly, no significantly modulated SGBs were identified in the placebo group throughout the intervention, which could be a result of differences in the sample size between Probio-M8 and placebo groups for the metagenomics analysis. The small sample size of the placebo group ($n = 17$ at the end of the intervention) also provided comparatively lower statistical

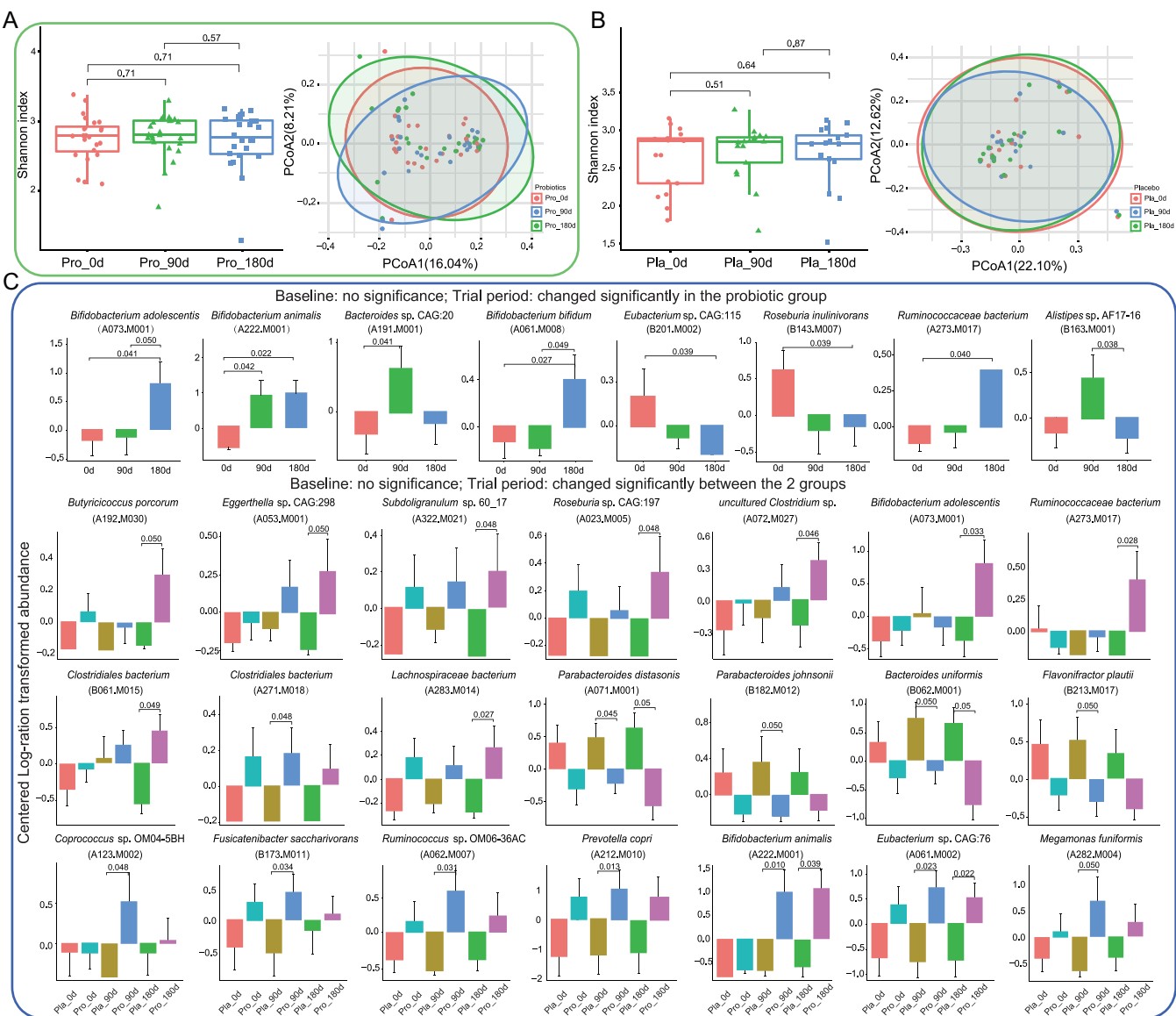

**FIG 2** Microbial diversity and species-level genome bins (SGBs) features of fecal metagenome data set of patients. Shannon diversity index and principal coordinates analysis (PCoA) score plots of the probiotic (A) and placebo (B) groups at days 0 (0d), 90 (90d), and 180 (180d). Symbols representing samples of the probiotic and placebo groups at different time points are shown in different colors. (C) Dramatic changed SGBs between probiotic and placebo groups at different time points. A total of 41 patients (probiotic group, $n = 24$; placebo group, $n = 17$) donated fecal samples at three consecutive time points for fecal metagenomics analysis. Statistical analysis was performed with the Wilcoxon test with correction for multiple testing using the Benjamini-Hochberg procedure and corrected $P < 0.05$ was considered statistically significant.

power than the Probio-M8 group ($n = 24$). Twenty-one differentially abundant SGBs were detected between the two groups at the end of the intervention. Compared with the placebo group, the abundance of 17 SGBs in the Probio-M8 group increased significantly, including *Butyricicoccus porcorum* (A192.M030), *Eggerthella* sp. CAG:298 (A053.M001), and *B. adolescentis* (A073.M001), while an opposite trend was observed in *Parabacteroides distasonis* (A071.M001), *P. johnsonii* (B182.M012), *Bacteroides uniformis* (B062.M001), and *Flavonifractor plautii* (B213.M017; Fig. 2C).

**Effect size and multivariable association analysis of gut microbiota and clinical features.** To reveal how intervention-induced gut microbiota alterations impacted the clinical features and the interactive association between the two during the intervention, an effect size (calculated by Adonis analysis) and multivariable association analysis were performed. Our analysis found that the gut microbiota of the Probio-M8 group explained a much larger variance in the clinical indices compared with the placebo group (Probio-M8

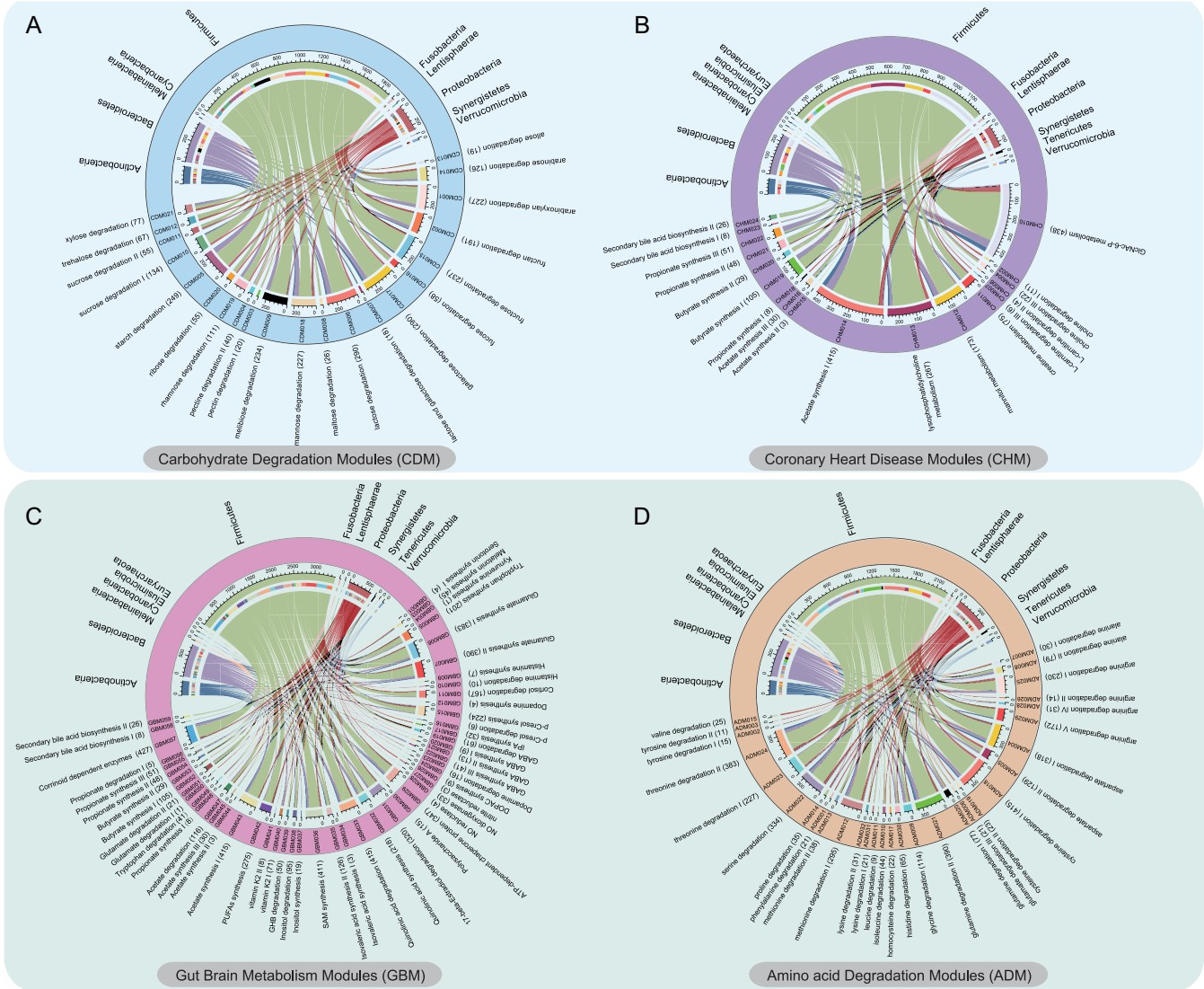

**FIG 3** A genome-centric metabolic reconstruction of patients with coronary artery disease was constructed. Four relevant module groups were analyzed, including (A) carbohydrate degradation modules (CDM), (B) coronary heart disease modules (CHM), (C) gut-brain metabolism modules (GBM), and (D) amino acid degradation modules (ADM). The upper part of the chord diagram shows the distribution of the species-level genome bins (SGBs) that encoded these modules across phyla. The lower part of the chord diagram represents the coded modules. The number of corresponding SGBs is given in parentheses.

0.458; placebo 0.041), corresponding to 23 and four significant effects contributing SGBs, respectively (Fig. S1B). In addition, 55 SGBs were found to be significantly related to the clinical indicators of the patients, six of which were also previously identified responsive SGBs, namely, uncultured *Clostridium* sp., *Roseburia inulinivorans*, *B. adolescentis*, *Eubacterium* sp. CAG:115, *Clostridiales* bacterium, and *B. animalis*. Some interesting correlations were found among these SGBs, and some investigated clinical features: the significant negative association between *B. animalis*, *Clostridium* sp. with LDL and SDS; significant positive association between *Eubacterium* sp. CAG:115, *B. adolescentis* with LDL and PL. *Roseburia inulinivorans* correlated significantly and positively with IL but negatively with PL (Fig. S1C). These results together suggested coadministering probiotics exerted stronger effects on the clinical features than conventional drug treatment alone by inducing changes in specific species.

**Adjunctive probiotics administration modulated gut microbiota-encoded metabolic modules.** A genome-centric metabolic reconstruction was built to discern changes in the GMMs encoded in the 440 SGBs using the MetaCyc and KEGG databases. With the consideration of target clinical phenotypes, we mainly included four module groups, namely, carbohydrate degradation (CDM, 21 modules; Fig. 3A), coronary heart disease (CHM, 25

modules; Fig. 3B), gut-brain metabolism (GBM, 60 modules; Fig. 3C), and amino acid degradation (ADM, 32 modules; Fig. 3D). Our results showed that 121 of these 138 included modules were effectively encoded by SGBs. The identified modules belonged to 13 phyla, distributed mainly to Firmicutes (70.68%), Bacteroidetes (11.59%), and Proteobacteria (7.95%). Galactose degradation (65.91%), lactose degradation (65.91%), *N-acetyl-D-glucosamine 6-phosphate* metabolism (99.55%), corrinoid dependent enzymes (97.05%), and cysteine degradation I (94.32%) were the most common encoded modules within the four module groups (Table S6). Notably, metabolic pathways related to dietary choline were mainly encoded by modules related to choline and carnitine degradation in the CHM module group, represented by 34 SGBs belonging mostly to Firmicutes (44.12%) and Proteobacteria (41.18%). The GBM module group encoded several short-chain fatty acids (SCFAs)-related metabolic modules, which were encoded by highly diverse SGBs. Specifically, 426, 94, and 167 SGBs were involved in acetate synthesis I, gamma-aminobutyric acid (GABA) metabolism, and cortisol degradation, respectively. Interestingly, there was no significant difference in the cumulative abundance of the four module groups (CDM, CHM, GBM, ADM) at 0, 90, and 180 days between Probio-M8 and placebo groups (Fig. S2).

**Adjunctive probiotics administration modulated gut microbiota-related bioactive compounds.** To identify probiotic-specific modulation of intestinal bioactive compounds, GMMs encoded by differential abundant SGBs between Probio-M8 and placebo groups were analyzed (Fig. 4A). Our results showed that the probiotic-receivers had more diverse SGBs participating in tryptophan synthesis and GABA metabolism during/after the intervention. Interestingly, four GBMs (including cortisol degradation, acetate, isovaleric acid, and polyunsaturated fatty acid [PUFAs] synthesis) were dominated in the Probio-M8 group, which were also characterized by a higher SGBs diversity and abundance in these modules, while the GBM module of acetate degradation was highly represented in the placebo group. In contrast, two modules of the CHM group, N-acetyl-D-glucosamine 6-phosphate (GlcNAc-6-P) and mannitol metabolism were dominated in the placebo group. Additionally, more choline degradation III-encoding SGBs were detected in the placebo group at 180 days. Among the ADM group-encoding SGBs, modules related to methionine, cysteine, glutamine, serine, and threonine degradation were dominated in the Probio-M8 group, while the placebo-receivers had more diverse SGBs participating in histidine synthesis and higher abundance of modules involved in aspartate degradation. It is also interesting to note that more SGBs in the Probio-M8 group were involved in arginine degradation V, contrasting to a higher proportion of arginine degradation I and IV involving SGBs in the placebo group. We also noted that during/after the intervention, probiotic-receivers had more diverse SGBs participating in sucrose degradation I and arabinose degradation, while the placebo group had a higher abundance of modules involved in lactose degradation, mainly encoded by two species, *Bacteroides uniformis* and *P. johnsonii*. Modules of starch, melibiose, and fructose degradation were dominated in the Probio-M8 group (characterized by a higher SGBs diversity and module abundance), while the xylose degradation module in the carbohydrate degradation module group was highly represented in the placebo group (Fig. 4A).

**Adjunctive probiotics administration modulated the predicted gut metabolome.** The gut bioactive metabolites of patients were identified through the MelonnPan pipeline, and a total of 80 metabolites were annotated. Eight representative metabolites showed nonsignificant differences between the placebo and Probio-M8 groups at day 0, and they became significantly differential features during/after the intervention (Table S7). Compared with the placebo group, the average predicted abundances of X3 methylxanthine, cytosine, malonate, and palmitoyl glycerol increased significantly in the probiotic-receivers, while C20:4 carnitine decreased significantly in the same group of subjects ($P < 0.05$; Fig. 4B), suggesting that coadministrating Probio-M8 with conventional drugs led to specific changes in some predicted metabolites, which reflected differences in the gut microbiome's potential to synthesize these compounds.

**Adjunctive probiotics administration modulated serum metabolites.** To identify disease-related changes and potential biomarkers for CAD, we then quantified the

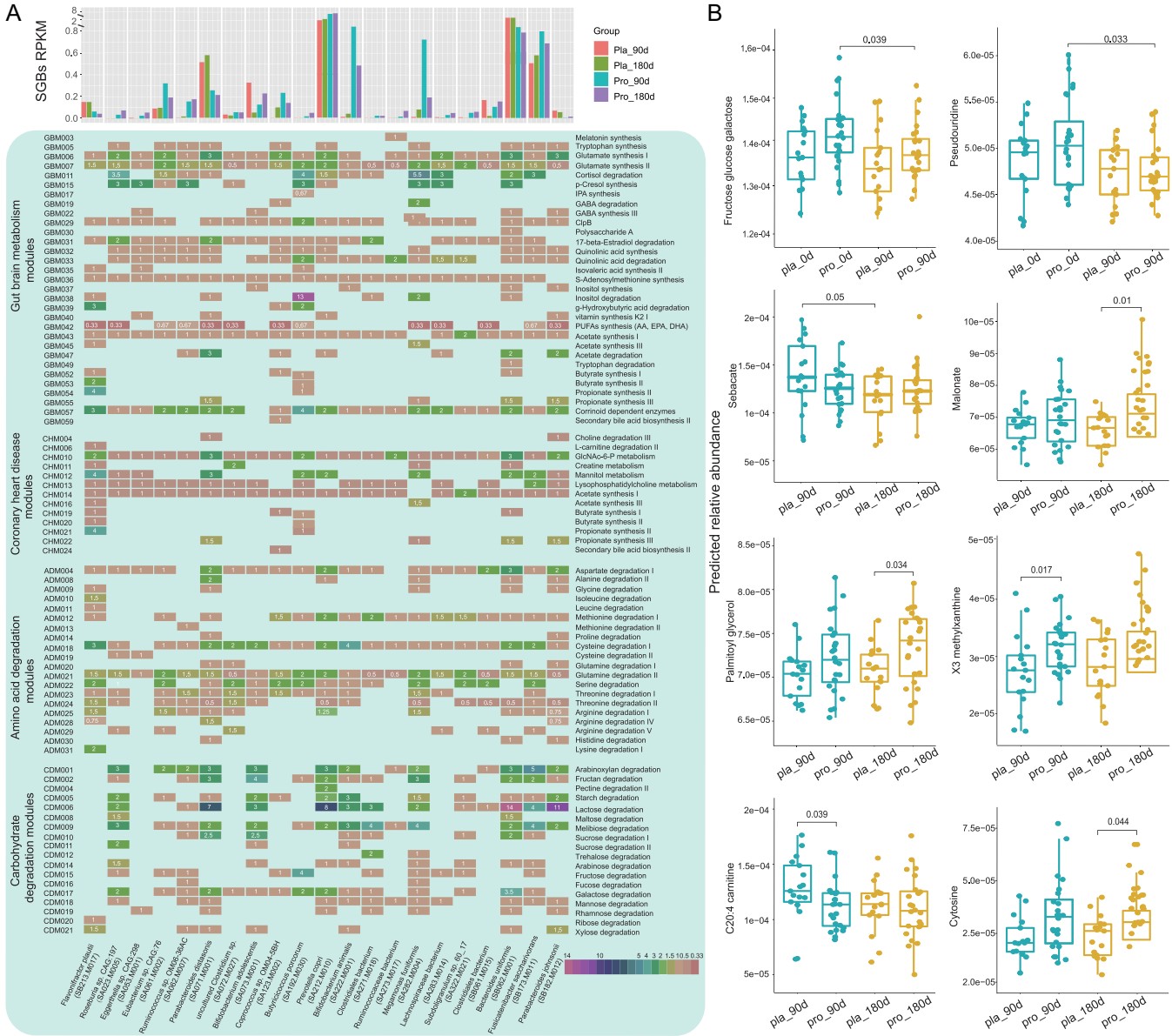

**FIG 4** Changes in the profiles of gut metabolic modules and predicted metabolites between the probiotic and placebo groups at different time points. (A) The histogram represents the abundance of identified responsive SGBs at different time points between the two groups (probiotic group, n = 24; placebo group, n = 17 in this analysis). The lower panel shows the distribution of components in four selected modules (carbohydrate degradation, coronary heart disease, gut-brain metabolism, and amino acid degradation) across the significantly differential species-level genome bins (SGBs) between the probiotic (Pro) and the placebo (Pla) groups. The square represents the presence of the specific modules in the corresponding SGB, and the number in the square represents the abundance of the module. (B) Boxplots showing the content of predicted differential bioactive metabolites that were responsive to the Probio-M8 adjuvant treatment. Statistical differences were evaluated with the Wilcoxon test. Benjamini-Hochberg procedure was used to correct for multiple testing and corrected *P* < 0.05 was considered statistically significant.

levels of 25 serum metabolites (including TMA, TMAO, and 23 amino acids) using targeted metabolomics (Table S8). Our results showed that Probio-M8 administration decreased both the serum levels of TMA (especially at day 90; *P* < 0.01) and TMAO (day 180 versus days 0 and 90; *P* < 0.05 in each case). In addition, significantly more TMAO was detected in the placebo group than the Probio-M8 group after 180 days of intervention (*P* < 0.001). Compared with the placebo group, the serum levels of two amino acids (L-serine and L-glycine) increased significantly while four other amino acids (L-leucine, L-valine, L-cysteine, and L-arginine) declined significantly in the Probio-M8 group (*P* < 0.05). The serum asparagine level increased significantly in the probiotic receivers, while the serum kynurenine level increased significantly in the placebo

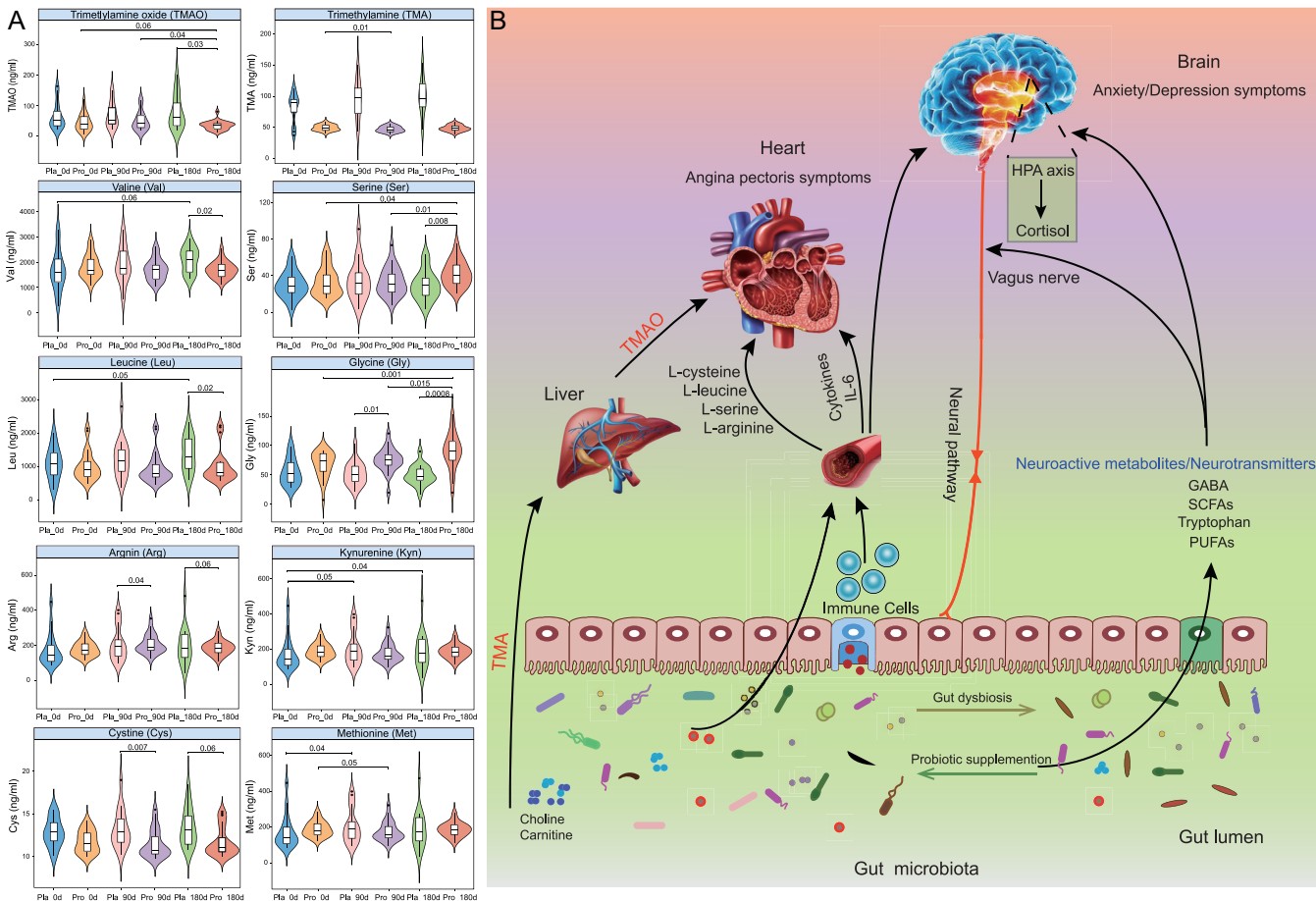

**FIG 5** Changes in serum metabolomes and proposed model of probiotic-driven pathways modulating the gut-heart/-brain axes in patients with coronary artery disease. (A) Violin plots showing the levels of serum metabolites (trimethylamine, trimethylamine-N-oxide, and specific amino acids) that were responsive to the Probio-M8 adjuvant treatment. A total of 41 patients provided blood samples at three consecutive time points for analysis of changes in serum metabolite levels (probiotic group, n = 24; placebo group, n = 17). Statistical differences were assessed by the Wilcoxon test or t test. Benjamini-Hochberg procedure was used for correction for multiple testing, corrected P < 0.05 was considered statistically significant. (B) Schematic diagram illustrating key probiotic-driven pathways that modulated the gut-heart/-brain axes and host response.

group ($P < 0.05$). Finally, compared with day 0, the serum level of ʟ-methionine was significantly reduced in the probiotic group but increased in the placebo group at day 90 ($P < 0.05$; Fig. 5A).

## DISCUSSION

The gut microbiota has been shown to affect the host physiological processes, and, in some cases, it even drives the development of disease (such as inflammatory bowel disease, asthma, anxiety, obesity, and CVD) through absorption of gut microbe-originated bioactive metabolites into the cardiovascular system (22, 23). The gut-heart axis is a recently established concept in the field of CVD research, and the bidirectional interactions between the two body locations may open innovative and promising diagnostic and treatment options for CAD (1, 3). This study investigated the added beneficial effects and mechanisms of applying the probiotic strain, Probio-M8, in conjunction with conventional regimen (atorvastatin and metoprolol) for CAD management.

The conventional regimen significantly increased the total scores of SAQ during/after the intervention, suggesting that the conventional regimen alone was enough to improve CAD-associated symptoms. However, the SAQ scores of patients receiving adjunctive Probio-M8 treatment were significantly higher than those given only the conventional regimen, especially in four subcategories, i.e., AF, AS, PL, and TS. Moreover, coadministering Probio-M8 significantly alleviated patients' anxiety and depression

symptoms. Meanwhile, the serum levels of IL-6 and LDL-C were significantly lower in the Probio-M8 group than the placebo group receiving only the conventional regimen. The significant differences in these clinical indicators between the two groups strongly suggested that probiotic adjuvant therapy synergized the clinical effects of conventional drug treatment.

Significant beneficial effects of applying probiotics have been reported in several previous studies. For example, a randomized clinical trial found that coadministering *L. rhamnosus* (LGG strain) and inulin for 8 weeks alleviated depression and anxiety and lowered the levels of inflammatory biomarkers in patients with CAD (24). Another study reported that ingesting multispecies probiotics for 12-week effectively inhibited the inflammation process, reduced homocysteine concentration, and improved serum lipid profile, altogether reducing the cardiometabolic risk (25). Malik et al. (26) reported that consuming *L. plantarum* 299v improved vascular endothelial function and decreased systemic inflammation in male subjects with CAD, though no significant effect was observed in the plasma TMAO level. However, it is worth noting that the clinical effects of probiotic intervention reported in human trials vary greatly, which could be a result of multiple factors, especially the study design. It is also known that the beneficial effects of probiotics are both strain-specific and host-specific. Thus, the beneficial effects of specific strains need to be investigated by independent clinical trials on a specified cohort of subjects under defined conditions. Data obtained in this study serve as valuable reference information supporting that Probio-M8 could be a potential strain to be further investigated for its clinical effects on anxiety, depression, and CAD symptoms.

The current work applied microbial metagenomic sequencing and in-depth bioinformatics analysis to provide a wealth of information on CVD-associated gut microbiota signatures (27, 28). No significant difference was found in the Shannon index at any time point between Probio-M8 and placebo groups. The results of PCoA analysis (Bray-Curtis distance) also did not exhibit any specific group/subgroup-based clustering pattern. The results of the alpha and beta diversity analyses suggested that the symptom alleviation effects seen in this work were not due to drastic shifts in the gut microbiota diversity but significant changes in the abundance of some SGBs. Some potentially beneficial and anti-inflammatory SGBs, e.g., *B. adolescentis*, *B. animalis*, *Butyricicoccus porcorum*, *Lachnospiraceae* bacterium, *Ruminococcaceae* bacterium, *Subdoligranulum* sp. 60_17, *Eubacterium* sp. CAG:76, and *Roseburia* sp. CAG:197, significantly increased after the intervention, while SGBs representing the species, *P. johnsonii*, *Bacteroides uniformis*, and *F. plautii*, decreased significantly. Other studies have also observed that alterations in some specific gut bacteria were closely related to the occurrence and progression of CAD. For instance, a recent study detected increased abundances of *Escherichia* and *Enterococcus*, accompanied by the reduction in the intestinal *Fecalibacterium*, *Roseburia*, and *Eubacterium* patients with CAD compared to the control group (29). Moreover, the relative abundance of *Lachnospiraceae* bacteria and *Ruminococcus gauvreauii* decreased significantly in patients with advanced CAD, while *Ruminococcus gnavus* increased significantly (30). A 12-week randomized, double-blind, placebo-controlled trial found that supplementing LGG significantly reduced the serum levels of total cholesterol and LDL-C in 44 patients with CAD (31). *Roseburia*, *Eubacterium*, and *Lachnospiraceae* bacteria are important butyrate-producers, which could significantly increase the cecum butyrate level, and butyrate serves as a signaling molecule that binds to specific receptors to regulate intestinal homeostasis and other essential physiological function across the endocrine system, nervous system, immune system, and cardiovascular system (32). These results indicated that maintaining intestinal health not only depends on the overall gut microbiota but also on microbes possessing the required functional capacity.

The modulation of the gut microbiota composition would simultaneously change the content of gut metabolites, some of which would be absorbed and transported to and through the blood circulation system, exerting biological effects on the host (33). Our results further showed that coadministering Probio-M8 enhanced microbial bioactive potentials modulated certain metabolic modules and regulated the serum

metabolome. The Probio-M8 receivers had more diverse SGBs participating in tryptophan synthesis and GABA metabolism during/after the intervention. As an intercellular signal, tryptophan metabolites can regulate intestinal inflammation, neuronal function, and mucosal immunity (34). Studies have shown that some gut microbes can produce GABA, which is the principal inhibitory neurotransmitter. It plays a key role in human anxiety and depression disorders via regulating the gut-brain axis (35). Our data showed that the abundance of species involved in GABA synthesis (such as *Bifidobacterium*, *Megamonas*, and *Eggerthella* spp.) was significantly higher in Probio-M8 receivers than those in the placebo group during/after the intervention, especially *B. adolescentis*. Interestingly, our results showed that the Probio-M8 group was dominated by SGBs encoding modules of cortisol degradation, acetate, isovaleric acid, and PUFAs synthesis, while acetate degradation was highly represented in the placebo group. Cortisol is the primary stress hormone associated with increased anxiety or depression. *Bifidobacterium* and *Lactobacillus* have recently been recognized as alternative treatments for anxiety and depression-like behaviors (36). Our results showed that Probio-M8 supplementation but not placebo could effectively relieve depression and anxiety symptoms in patients. Multivariate association analysis also confirmed that the abundance of *B. animalis* correlated significantly and negatively with depressive symptoms. A study reported that ingesting the probiotic mix, *L. helveticus* R0052 and *B. longum* R0175, for 1 month could improve depression and anxiety while reducing the cortisol level in humans (37). Another group of neuroactive molecules, SCFAs, especially acetate, butyrate, propionate, could directly/indirectly participate in communication along the gut-brain axis (38). Probio-M8 co-supplementation could improve anxiety and depression symptoms in patients, and such effect was likely achieved by conveying molecular signals to the endocrine, vagal, and intestinal nervous system through the gut-brain axis (Fig. 5B).

The gut microbiome can act as a virtual endocrine system to regulate distal organs through metabolism-dependent pathways and biomolecules like TMA/TMAO, SCFAs, and bile acids (39). One gut microbiota-originated metabolite is TMAO, which has attracted much attention due to its role in the onset and promotion of atherosclerosis and cardiometabolic diseases (40). Dietary nutrients that contain a TMA moiety (such as choline, phosphatidylcholine, and L-carnitine) are processed by bacterial TMA-lyases to form TMA, and the released TMA is transported to the liver after absorption, where the hepatic enzyme flavin monooxygenase-3 would oxidize TMA to TMAO (41). In this study, the placebo group had significant increases in the predicted levels of C20:4 carnitine at day 90 and SGBs involved in choline degradation III at day 180. In contrast, Probio-M8 administration significantly reduced the serum TMA and TMAO levels compared to baseline. More importantly, the TMAO level of patients in the Probio-M8 group was significantly lower than that in the placebo group after intervention. A previous meta- and dose-response analysis reported that elevated plasma TMAO concentrations were associated with an increased incidence of major adverse cardiovascular events in patients with CAD (42). Our finding corroborated the observation of Yao et al. (42) that Probio-M8 intervention reduced the level of TMAO, accompanied by a decrease in the incidence of adverse cardiovascular events in patients with CAD. Some other previous studies also observed similar beneficial effects conferred by probiotic consumption. For example, ingesting fermented foods that contained the probiotic strains, *L. rhamnosus* LRH11 and *L. plantarum* SGL07, continuously for 16 weeks could effectively reduce serum TMAO levels in subjects at risk of CVD (43). Another example was the probiotic *B. animalis* subsp. *lactis* LKM512, which has been shown to reduce the risk of arteriosclerosis development in healthy subjects via reducing the release of gut-microbiota-originated TMA (44). These results consistently supported that regular intake of probiotics could lower the risk for CVD, and one possible protection mechanism was via reducing TMA and TMAO concentrations (Fig. 5B).

Some amino acids have been identified as novel biomarkers and metabolic signatures for CVD (45). Branched-chain amino acids (i.e., leucine, isoleucine, and valine) have received considerable attention due to their consistently high plasma levels in

individuals at risk of CVD (46). Our study found that the Probio-M8 group had significantly lower serum levels of L-leucine, L-valine, L-cysteine, and L-arginine. A previous study showed that the baseline leucine or isoleucine concentrations were significantly associated with CVD risk after adjusting for potential confounders, and such correlation was even stronger in cardiovascular high-risk groups (47).

In addition to branched-chain amino acids, several other amino acids, including methionine, cysteine, and arginine, have been found to promote the development of atherosclerosis in human and animal models via different mechanisms, e.g., impairing antioxidant activity, increasing lipid peroxidation, and through enhancing macrophage foam cell formation (46, 48). Probio-M8 supplementation significantly increased subjects' serum L-serine and L-glycine levels. L-serine is considered an antioxidant and cytoprotective molecule because it could stimulate certain key antioxidant factors (49). Glycine correlates negatively with the risk of acute myocardial infarction in patients with suspected angina pectoris, reducing the chance of CVD and anti-atherosclerosis (50). Thus, the increase in serum concentrations of these amino acids after Probio-M8 consumption could be part of the protective mechanism against CVD (51, 52).

In conclusion, this study demonstrated that coadministering Probio-M8 with a conventional regimen offered added benefits to patients with CVD compared with conventional treatment alone. The beneficial effects of Probio-M8 intake were likely achieved through influencing the gut microbiome and metabolomes, which in turn modulated multiple pathways of both gut-heart and gut-brain axes, e.g., via increasing the diversity and abundance of anti-inflammatory gut microbial species, decreasing the serum levels of TMAO/TMA, and modulating the levels of specific amino acid and bioactive metabolites. Meanwhile, the corresponding predicted metagenomic metabolism modules were also regulated by Probio-M8 consumption. Our data supported that probiotic treatment synergized with conventional therapy to alleviate diseases associated with the gut-heart axis like CAD, expanding the management options for this spectrum of disease.

## MATERIALS AND METHODS

**Experimental design and subject recruitment.** This study was a 6-month randomized double-blinded placebo-controlled clinical trial. Eighty CVD patients were recruited from the Medical Clinic of Weihai Municipal Hospital (Weihai City, Shandong Province, China). All patients admitted to this hospital with a diagnosis of CAD were considered for participation in the study and screened by a cardiologist for eligibility. The inclusion criteria were as follows: male or female with diagnostic criteria for CAD; all patients expressed willingness to commit throughout the trial. Patients were excluded if they had: acute coronary syndrome; a history of major diseases, such as gastrointestinal diseases, mental illness, type I diabetes; taken antibiotics, probiotics, prebiotics, postbiotics, and immunosuppressive agents within 1 month before the intervention or during the intervention; taken statins before hospitalization; irregular eating habits during the intervention and follow-up period.

Nine subjects were excluded after the first round of screening, 70-one patients were randomly assigned to probiotic (n = 37) and placebo (n = 34) groups. After being informed of the specific experimental guidelines and details, six patients withdrew from the trial; five patients were transferred to other hospitals, and 36 and 24 subjects remained in the probiotic and placebo groups subsequently (Table S1). Both participants and researchers were unaware of the group allocation throughout the trial. The probiotic group received both atorvastatin, metoprolol, and one sachet of Probio-M8 powder per day (two grams per sachet; $3 \times 10^{10}$ CFU/sachet/day), while the placebo group received atorvastatin, metoprolol, and placebo material. All patients received a daily dose of 20 mg of atorvastatin in the form of a tablet, which was taken before bedtime. The intervention continued for 6 months. The Probio-M8 powder and placebo material were manufactured by Jinhua Yinhe Biological Technology Co., Ltd., China. The sachets of both probiotic and placebo materials appeared as a light pink powder and were identical in weight, taste, and appearance.

Self-administered questionnaires and patients' clinical indicators, including SAQ, SAS, SDS, RBT, IL-6, LDL-C, cereal third transaminase, blood urea nitrogen, and creatinine, were recorded at days 0 and 180. Fecal and blood samples were collected at days 0, 90, and 180. However, due to the withdrawal of patients and the failure of continuous donation of fecal and blood samples at all three time points, only 24 and 17 patients remained in the probiotic and placebo groups, respectively, for the follow-up metagenomics and metabonomic analyses (Fig. S1A).

**Shotgun sequencing, contig binning, genome dereplication.** The procedures for total fecal DNA extraction, sequencing, and construction of sequencing libraries were described in our previous works (13). A total of 123 stool samples were shotgun sequenced (n = 24 and 17 for Probio-M8 and placebo groups, respectively; sampled at days 0, 90, and 180 for each individual).

Reads of each sample were assembled into contigs using MEGAHIT (53). Contigs greater than 2,000 bp were selected for binning using MetaBAT2, VAMB, and DAS Tool with default options (54, 55). Then, the results of the three binners were combined to obtain metagenome-assembled genomes (MAGs) using in-house scripts. Sample reads were mapped back to the corresponding contigs using BWA-MEM2 (56), and the contig depth was calculated using Samtools and the jgi_summarize_bam_contig_depths function in MetaBAT2 (57).

The completeness and contamination of MAGs were evaluated using CheckM ([https://github.com/Ecogenomics/CheckM](https://github.com/Ecogenomics/CheckM)), and they were classified as high-quality (completeness $\geq$ 80%, contamination $\leq$ 5%), medium-quality (completeness $\geq$ 70%, contamination $\leq$ 10%), and partial-quality (completeness $\geq$ 50%, contamination $\leq$ 5%) MAGs. The high-quality genomes were clustered, and the most representative genome from each replicate set was selected by dRep for the extraction of species-level genome bins (SGBs), using the parameters $-$pa 0.95 and $-$sa 0.95 (Fig. 1A).

**Taxonomic annotation and calculation of relative abundance of SGBs.** The SGBs were annotated through Kraken2 and the NCBI nonredundant Nucleotide Sequence Database (retrieved in November 2020), and the predicted genes were searched against the UniProt Knowledgebase (UniProtKB, released 2020.11) using the blastp function of DIAMON with default options. The relative abundance of each SGBs was calculated through coverM ([https://github.com/wwood/CoverM](https://github.com/wwood/CoverM)) using the parameter "--min-read-percent-identity 0.95 --min-covered-fraction 0.4".

**Prediction of relevant gut metabolic modules (GMMs) and bioactive metabolites.** The published literature and MetaCyc metabolic database were used to predict SGBs encoding relevant GMMs (58–61). For each SGB, the predicted open reading frames (ORFs) were compared with the Kyoto Encyclopedia of Genes and Genomes (KEGG) Orthologues (KOs) database to annotate the key metabolic modules. Profiles of SGBs encoding synthesis and/or degradation-related modules were identified by Omixer-RPM using the parameter $-$c 0.66 (62).

The profiles of gut bioactive metabolites were predicted. First, one million reads per sample were extracted using seqtk ([https://github.com/lh3/seqtk](https://github.com/lh3/seqtk)), and these sequences were compared using the blastx function of DIAMON with the parameters –query-cover 90 –id 50. The gene abundance profile of each sample was then calculated by the best hit of each gene. Finally, the MelonnPan-predict pipeline was applied to convert the gene abundance profiles into predicted bioactive metabolites profiles (63).

**Determination of serum metabolites by liquid chromatography-mass spectrometry (LC-MS).** Several substrates, including trimethylamine (TMA), TMAO, and specific amino acids, were detected in patients' serum samples using the QTRAP 6500+ UPLC-MS/MS System (SCIEX Co., USA). The serum metabolite extraction, sample preparation, and quantification methods were a reference to Wu et al. (64) with some modifications. Briefly, all serum samples were centrifuged at 10,000 g for 10 min to remove the impurity, and then 100 $\mu$L of samples were mixed with 400 $\mu$L of acetonitrile for 30 s. The mixtures were centrifuged at 10,000 $\times$ g for 10 min to precipitate proteins. The supernatants were filtered through a 0.22 $\mu$m syringe filter, and 200 $\mu$L of each supernatant was used for ultra-performance liquid chromatography-tandem mass spectrometry (UPLC-MS/MS) analysis. All the prepared solutions were stored in the dark at 4℃ until use.

**Conditions of UPLC-QTRAP-MS/MS.** Chromatographic separation was carried out on an ACQUITY UPLC HSS T3 column (100*2.1 mm, 1.8 $\mu$m, Waters, Ireland). The flow rate was maintained at 0.3 mL/min, and the column was heated to 40℃. The mobile phase A was a mixture of water and acetonitrile (19:1) containing 0.1% formic acid, and mobile phase B was acetonitrile containing 0.1% formic acid. The gradient conditions were: 0 to 3 min 100% A,3 to 10 min100 to 25% A,10 to 10.5 min25 to 0% A,10.5 to 16.5 min 0% A (flow rate: 0.6 mL/min),16.5 to 17 min 0 to 100% A,17 to 20 min 100% A. The parameters of QTRAP 6500+ mass spectrometry were: curtain gas, 35 lb/in$^2$; collision activation dissociation gas, medium; ion source gas 1, 50 lb/in$^2$; the ion spray voltage of positive mode, 5,500 V; and the ion spray voltage of negative mode, 4,500 V. The information and metabolites detected by multiple reaction monitoring-ion pair channel and the corresponding declustering potential and collision energy were listed in Table S2.

**Statistical analyses.** The multiomics analyses pipeline is shown in Fig. 1A, and graphical presentations were generated using the R software (v.4.0.2) and Adobe Illustrator. Principal coordinates analysis (PCoA) was performed and visualized using the R package vegan and ggpubr, while the adonis $P$ value was generated based on 999 permutations. The Wilcoxon test and the $t$ test were used to evaluate differences in various variables between groups. $P$ values were corrected for multiple testing using the Benjamini-Hochberg procedure. The effect size analysis was performed according to Yan et al. (22). Briefly, a set of nonredundant covariates (species versus clinical indicators, metabolites, or immune parameters) were selected from the omics and clinical parameter data sets, and then the accumulated effect size was calculated by Adonis analysis in each case. Multivariate associations between species features and clinical indicators were performed by the R package, MaAsLin2. Centered log-ratio (clr) abundance transformation was performed using the microbiome R package. The vegan package was used to determine the similarity between two multivariate axes (999 permutations). Cumulative abundances of modules were calculated by the dplyr package in R, using the formula: module accumulated abundance = number of metabolic modules encoded in genomes $\times$ genome abundance.

**Data availability.** Sequencing data generated in this study and custom analysis codes are available in NCBI-SRA (BioProject: PRJNA741822) and under [https://github.com/TengMa-Cleap/Probiotics-relieve-coronary-heart-disease-project](https://github.com/TengMa-Cleap/Probiotics-relieve-coronary-heart-disease-project), respectively.

## SUPPLEMENTAL MATERIAL

Supplemental material is available online only.

**FIG S1**, PDF file, 0.2 MB.

**FIG S2**, PDF file, 0.9 MB.
**TABLE S1**, PDF file, 0.1 MB.
**TABLE S2**, PDF file, 0.1 MB.
**TABLE S3**, PDF file, 0.1 MB.
**TABLE S4**, PDF file, 0.1 MB.
**TABLE S5**, PDF file, 0.1 MB.
**TABLE S6**, PDF file, 0.1 MB.
**TABLE S7**, PDF file, 0.1 MB.
**TABLE S8**, PDF file, 0.1 MB.

## ACKNOWLEDGMENTS

This research was supported by the National Natural Science Foundation of China (grant number 31720103911), the China Agriculture Research System of MOF and MARA, and the Science and Technology Major Projects of Inner Mongolia Autonomous Region (grant number 2021ZD0014).

The study was approved by the Ethics Committee of the Weihai Municipal Hospital (project number 201816) and was registered on the Chinese Clinical Trial Registry (http://www.chictr.org.cn/; registration number ChiCTR1800017162). The investigation was conformed to the principles outlined in the Declaration of Helsinki. Informed consent was obtained from all recruited subjects before the study.

Zhihong Sun and Heping Zhang: conceptualization, design of methodology. Baoqing Sun: clinical trial implementation, specimen collection. Teng Ma: formal analysis, data curation, visualization, writing of the original draft. Xinfu Zhou: patient recruitment. Yalin Li: formal analysis, software testing, and verification. Bohao Li, Shuai Guo, and Ni Yang: sample processing, conducting experimental work. Lai-Yu Kwok: writing, critical evaluation, and revision of the original draft, resource provision. Shukun Zhang: supervision of the clinical trial, manuscript revision.

We declare no conflict of interest.

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
