## [Reviewer comments · mSystems]

Bifidobacterium lactis Probio-M8 adjuvant treatment confers added benefits to patients with coronary artery disease via target modulation of the gut-heart/-brain axes

Baoqing Sun, Teng Ma, Yalin Li, Ni Yang, bohaili, Xinfu Zhou, Shuai Guo, Shukun Zhang, Lai-Yu Kwok, zhihong sun, and Heping Zhang

Corresponding Author(s): Heping Zhang, Inner Mongolia Agricultural University

Review Timeline:

Submission Date:	January 31, 2022
Editorial Decision:	February 14, 2022
Revision Received:	February 15, 2022
Accepted:	February 16, 2022

Editor: Sean Gibbons

Reviewer(s): Disclosure of reviewer identity is with reference to reviewer comments included in decision letter(s). The following individuals involved in review of your submission have agreed to reveal their identity: Tomasz Wilmanski (Reviewer #1)

Transaction Report:

DOI: <https://doi.org/10.1128/msystems.00100-22>

February 14, 2022

Dr. Teng Ma
Inner Mongolia Agricultural University
Hohhot
China

Re: mSystems00100-22 (Bifidobacterium lactis Probio-M8 adjuvant treatment confers added benefits to patients with coronary artery disease via target modulation of the gut-heart/-brain axes)

Dear Dr. Teng Ma:

Thank you for submitting your manuscript to mSystems. We have completed our review and I am pleased to inform you that, in principle, we expect to accept it for publication in mSystems. However, acceptance will not be final until you have adequately addressed the reviewer comments. Specifically, Reviewer 1 had some additional concerns about your statistical analyses that need to be thoroughly addressed prior to resubmission.

Preparing Revision Guidelines

Sincerely,

Sean Gibbons

Editor, mSystems

Journals Department
American Society for Microbiology
1752 N St., NW

Reviewer comments:

Reviewer #1 (Comments for the Author):

The present study by Sun et al. investigated the effect of a probiotic intervention (Probio-M8, *Bifidobacterium lactis*) in combination with conventional therapy on CVD-related outcomes, microbiome composition, and metabolite levels in individuals diagnosed with CVD. This is a resubmission of a paper I have previously provided feedback on. The authors have addressed many of my previous concerns regarding sample size, typos in the text, and statistical analyses. Overall, the paper has improved considerably since the previous submission. However, several further comments and concerns regarding statistical analyses are provided below. Mainly there is some inconsistency in correcting for type 1 error, the appropriateness of the statistical tests performed, and some components that require further clarification.

1) In figures 1b, the description of the statistical analysis is very vague. Were the samples analyzed in a repeated measures design (paired samples t-test or paired Wilcoxon?). In addition, there is no mention of correction for multiple hypothesis testing. Are the p-values provided unadjusted p-values? A more appropriate approach would involve a repeated measures ANOVA, likely with a intervention-by-time interaction term included in the model, followed by multiple-group comparisons with a correction for type 1 error.

2) Along the same lines, given this was a randomized placebo-control trial, what were the primary outcomes defined a priori before initiating the trial? More information on these specific parts of the study design is required.

3) Lines 375-378: "Notably, results of Procrustes analysis confirmed a good cooperativity between the gut microbiome and metabolome (correlation=0.341; $P=0.001$; Figure 4b), revealing consistent changes between the gut SGBs and predicted metabolites during the intervention."

I am not sure I understand what this analysis shows. Metabolite abundances were inferred from the metagenomic content of the microbiome samples. Therefore, it seems redundant to show that these inferred metabolite abundances show a significant correspondence to the gut metagenome from which they were inferred. In Figure 4b Procrustes is also misspelled. Please correct.

4) Lines 309-313: "To reveal how intervention-induced gut microbiota alterations impacted the clinical features and the interactive association between the two during the course of intervention, a PERMANOVA-based effect size and multivariable association analysis was performed. Our analysis found that the gut microbiota of Probio-M8 group explained a much larger variance in the clinical indices compared with the placebo group (Probio-M8 0.458; placebo 0.041)".

It is unclear to me what analysis was performed here by the researchers. PERMANOVA classically models the dissimilarity matrix calculated from gut microbiome compositions as the dependent variable, testing the association with phenotypic features. There is no mention of PERMANOVA in the statistical methods, and the way the results are presented doesn't appear PERMANOVA was applied in the classical sense. Please explain this analysis further and in more detail.

5) Lines 383-384: "...suggesting that co-administrating Probio-M8 with conventional drugs led to changes in some of the predicted gut metabolites, particularly certain microbial bioactive compounds."

Since these metabolite differences are all inferred from gut metagenomics data, and not measured directly, the authors should clearly state that these metabolite changes at best reflect differences in the gut microbiome's potential to synthesize these compounds.

Minor comments:

Figure 5b has a duplication of the p-value bars. Please correct.

16 February 2022

Dear Dr. Gibbons,

Thank you for your and the reviewers' comments and suggestions on our manuscript. The comments and suggestions are valuable for improving our manuscript. We have read the comments carefully and revised accordingly, and the revised portion of the manuscript is shown in red in the updated manuscript.

We hope that our revised version will now be acceptable for publication in mSystems, and we look forward to hearing from you. Thank you for your time and consideration.

Best regards,

Teng Ma

Answers to reviewers:

Reviewer #1:

The present study by Sun et al. investigated the effect of a probiotic intervention (Probio-M8, *Bifidobacterium lactis*) in combination with conventional therapy on CVD-related outcomes, microbiome composition, and metabolite levels in individuals diagnosed with CVD. This is a resubmission of a paper I have previously provided feedback on.

The authors have addressed many of my previous concerns regarding sample size, typos in the text, and statistical analyses. Overall, the paper has improved considerably since the previous submission. However, several further comments and concerns regarding statistical analyses are provided below. Mainly there is some inconsistency in correcting for type 1 error, the appropriateness of the statistical tests performed, and some components that require further clarification.

Thank you for your time and new suggestions to us and our manuscript, which further have further improved the quality of this work. In the current new revised version, we have responded to all your comments and suggestions accordingly, and a point-by-point reply is provided below. Thank you again!

[1] In figures 1b, the description of the statistical analysis is very vague. Were the samples analyzed in a repeated measures design (paired samples t-test or paired Wilcoxon?). In addition, there is no mention of correction for multiple hypothesis testing. Are the p-values provided unadjusted p-values? A more appropriate approach would involve a repeated measures ANOVA, likely with a intervention-by-time interaction term included in the model, followed by multiple-group comparisons with a correction for type 1 error.

Answer: Thank you very much for your comment. The clinical indicators data were firstly assessed by the Shapiro-Wilk normality test (all *P*-values were > 0.05), indicating that the data were normally distributed. Thus, it would have been appropriate to use paired samples t-tests for evaluating the difference between the probiotic or placebo groups at different time points (e.g., pla_0d vs pla_180, pro_0d vs pro_180), and horizontal comparisons between probiotic and placebo groups (e.g., pla_0d vs pro_0d, pla_180d vs pro_180d) were assessed using Wilcoxon tests. All the provided p-values were corrected for multiple testing using the

Benjamini-Hochberg procedure, and corrected $P < 0.05$ was considered statistically significant. We have modified the figure legend to clarify the details of statistical analysis. Thank you again for your suggestion, which has greatly helped us improve the quality of our manuscript.

[2] Along the same lines, given this was a randomized placebo-control trial, what were the primary outcomes defined a priori before initiating the trial? More information on these specific parts of the study design is required.

Answer: Thank you very much for your comment. We specified the primary outcomes in the last paragraph of the Introduction section. The primary outcomes were various clinical indicators for coronary artery disease, namely angina frequency (AF), angina stability (AS), disease perception, physical limitation (PL), treatment satisfaction (TS), anxiety and depression levels (evaluated by the scores on the Self-Rating Anxiety Scale scores [SAS], and Self-Rating Depression Scale scores [SDS]); serum indicators, including interleukin-6 (IL-6), low-density lipoprotein cholesterol (LDL-C), cereal third transaminase, blood urea nitrogen, and creatinine; and white blood cell count. Additionally, changes in patients' fecal metagenomes and serum marker metabolites were followed. Please see line 117-124.

[3] Lines 375-378: "Notably, results of Procrustes analysis confirmed a good cooperativity between the gut microbiome and metabolome (correlation=0.341; $P=0.001$; Figure 4b), revealing consistent changes between the gut SGBs and predicted metabolites during the intervention."

I am not sure I understand what this analysis shows. Metabolite abundances were inferred from the metagenomic content of the microbiome samples. Therefore, it seems redundant to show that these inferred metabolite abundances show a significant correspondence to the gut metagenome from which they were inferred. In Figure 4b Procrustes is also misspelled. Please correct.

Answer: Thank you very much for your comment. We agree with the Reviewer's point of view. It is correct that metabolite abundances were inferred from the metagenomic content of the microbiome samples. Owing to the redundancy, the part was removed.

[4] Lines 309-313: "To reveal how intervention-induced gut microbiota alterations impacted the clinical features and the interactive association between the two during the course of intervention, a PERMANOVA-based effect size and multivariable association analysis was performed. Our analysis found that the gut microbiota of Probio-M8 group explained a much larger variance in the clinical indices compared with the placebo group (Probio-M8 0.458; placebo 0.041)". It is unclear to me what analysis was performed here by the researchers. PERMANOVA classically models the dissimilarity matrix calculated from gut microbiome compositions as the dependent variable, testing the association with phenotypic features. There is no mention of PERMANOVA in the statistical methods, and the way the results are presented doesn't appear PERMANOVA was applied in the classical sense. Please explain this analysis further and in more detail.

Answer: Thank you very much for your comment. We apologize for the mistake. It was indeed just "effect size analysis", and no PERMANOVA was performed. Correction was made accordingly. Briefly, the current method was carried out by inputting data of non-redundant covariates from the omics and parameter datasets (e.g., species vs metabolites, clinical indicators, or immune parameters), and then the accumulated effect size was calculated by Adonis analysis. This effect size analysis is currently widely used in multi-omics analysis. We have changed the manuscript and added further explanations in Methods. Please see line 232-235, and 310-311.

[5] Lines 383-384: "...suggesting that co-administrating Probio-M8 with conventional drugs led to changes in some of the predicted gut metabolites, particularly certain microbial bioactive compounds." Since these metabolite differences are all inferred from gut metagenomics data, and not measured directly, the authors should clearly state that these metabolite changes at best reflect differences in the gut microbiome's potential to synthesize these compounds.

Answer: Thank you very much for your comment. We strongly agree with you, and I have revised the manuscript here to "suggesting that co-administrating Probio-M8 with conventional drugs led to specific changes in some predicted metabolites, which reflected differences in the gut microbiome's potential to synthesize these compounds." Please see line

380-382.

[6] Figure 5b has a duplication of the p-value bars. Please correct.

Answer: Thank you very much for your comment. We carefully checked all the figures, and the duplication of p-value bars in Figure 2b was corrected.

February 16, 2022

Dr. Teng Ma
Inner Mongolia Agricultural University
Hohhot
China

Re: mSystems00100-22R1 (Bifidobacterium lactis Probio-M8 adjuvant treatment confers added benefits to patients with coronary artery disease via target modulation of the gut-heart/-brain axes)

Dear Dr. Teng Ma:

Your manuscript has been accepted, and I am forwarding it to the ASM Journals Department for publication. For your reference, ASM Journals' address is given below. Before it can be scheduled for publication, your manuscript will be checked by the mSystems production staff to make sure that all elements meet the technical requirements for publication. They will contact you if anything needs to be revised before copyediting and production can begin. Otherwise, you will be notified when your proofs are ready to be viewed.

Publication Fees:

We recognize that the video files can become quite large, and so to avoid quality loss ASM suggests sending the video file via <https://www.wetransfer.com/>. When you have a final version of the video and the still ready to share, please send it to mSystems staff at mSystemsjournal@msubmit.net.

For mSystems research articles, if you would like to submit an image for consideration as the Featured Image for an issue, please contact mSystems staff at mSystemsjournal@msubmit.net.

Sincerely,

Sean Gibbons